# Virtual Field Trips in Binational Collaborative Teacher Training: Opportunities and Challenges in the Context of Education for Sustainable Development

Caroline Leininger-Frézal [1,*] and Sandra Sprenger [2,*]

1   Department of History Geography and Social Science, Université Paris Cité, 75013 Paris, France
2   Department of Social Sciences, Mathematics and Natural Sciences Education, Universität Hamburg, 20146 Hamburg, Germany
*   Correspondence: caroline.leininger-frezal@u-paris.fr (C.L.-F.); sandra.sprenger@uni-hamburg.de (S.S.)

**Abstract:** Virtual field trips (VFTs) are a way to overcome travel restrictions and continue to explore distant spaces, foreign issues, and learning opportunities. The intent of this article is to show how VFTs are used in learning geography in the context of education for sustainable development (ESD). The goal is to develop a didactic approach to the use of virtual fieldwork in ESD with geography teachers in pre-service and in-service teacher training at two universities. This article has the following aims. The first is to explain what a VFT is from a theoretical and technical perspective, which raises questions about forms and tools. The second is to explain how a remote intercultural seminar was conceived and implemented to create virtual fieldwork. The third is to present the methodology on which this experimentation is based and to explore the opportunities and limitations of VFTs. The last is to present and discuss the results.

**Keywords:** virtual field trip; education for sustainable development; experiential learning; in-service and pre-service teacher training

## 1. Introduction

Virtual field trips (VFTs) offer students the opportunity to explore the world from the comfort of their homes or bring the living world into the classroom [1]. They are an exciting tool for private individuals and the educational sector alike. In education, particularly geography, field trips are a central teaching tool. They are used from kindergarten to the secondary and graduate levels and are components of many curricula [2,3]. As the whole world is coping with the COVID-19 pandemic, VFTs are becoming increasingly important. Through them, it is possible to visit places despite travel restrictions. However, VFTs are not only helpful during a pandemic. Sometimes, people are unable to travel for family, health, or financial reasons. Here, new opportunities arise [4]. Fieldwork is central to learning geography [1,2,4–9] in both tertiary and secondary education. However, real field trips have disadvantages because they can be costly in terms of both time and money, challenging to organize, and not necessarily accessible to people with disabilities [10]. These constraints are even more amplified during the current COVID-19 pandemic. A VFT is a way to overcome travel restrictions and continue to explore distant spaces, foreign issues, and learning opportunities. This article aims to develop a didactic approach to the use of virtual fieldwork in education for sustainable development (ESD), showing how virtual field trips are used in learning geography. Geography teachers in initial and continuing teacher training in two universities (Hamburg and Paris) participated in this project based on the 4Is approach, presented later in the text [11]. The following two theses are proposed:

- Thesis 1: The 4Is approach can be implemented in a VFT.
- Thesis 2: The 4Is approach allows students to understand the organization of the explored spaces and identify their sustainability issues.

This article is structured in five different sections: Section 2 explains what a VFT is from a theoretical and technical perspective, which raises further questions about forms and tools. Section 3 highlights the conceptualization of the seminar and how a remote seminar was conceived and implemented to create virtual fieldwork. Section 4 explains the methodology on which this experimentation is based. Section 5 presents the results and the opportunities and limitations of VFTs. Section 6 consists of the discussion.

## 2. Theoretical Background

### 2.1. Education for Sustainable Development

ESD is the underlying core educational concept of this study. This concept aims to provide students with knowledge about sustainable development (SD) and help them understand issues related to its environmental, social, and economic dimensions [12,13]. In recent years, many contributions have been made to ESD, especially in higher education, using different approaches. The implementation of ESD is conducted within the subject matter, as for geography education [14]. In this study, ESD is differentiated from the teaching training in the major of geography.

In the ESD context, the literature describes objectives and skills that students can develop. The competencies are very heterogeneous. They refer to ESD generally and without reference to a level of education [15,16]. In order to implement the dimensions of sustainability in the field of education, de Haan [15] developed the concept of "shaping competence", which consists of 12 sub-competencies. Examples of the sub-competencies are "integrating new perspectives", "think and act in a forward-looking manner", and "deal with incomplete and overly complex information" [15]. While the concept refers to educational processes in general, there are other approaches that refer to specific phases of the educational system. For example, Rieckmann focuses particularly on higher education [17]. There have been many approaches to ESD-related competencies in recent years. An overview can be found in the study by Rieckmann [16]. The competencies are also specified in relation to teacher training. One model of competencies central to teacher education consists of the 12 ESD teaching competencies that emerged from the framework A Rounder Sense of Purpose [18]. These competencies can be used as the basis of teacher training to improve capabilities in relation to ESD [18].

While ESD approaches are implemented in different ways in teacher education and geography education (e.g., [14,19]), this project is a transnational approach to the implementation of ESD in teacher training in geography. Sustainable Development Goal 11 concerning sustainable cities and communities [20] and its related objectives [21] are the focus of this approach.

### 2.2. (Virtual) Field Trips in Geography Education

Fieldwork is central to geography training at the university level [1,2,4–9]. One cannot become a geographer without doing fieldwork. It is in the field that students learn the attitudes and methodologies necessary to become geographers. Fieldwork constitutes an initiatory ritual. "The initiate gains access to the community because he in turn holds the gestures that constitute the identity of the group, which he acquires only through experience" (translated from [9]). Field trips and fieldwork have been implicit and polysemic for a long time. The term fieldwork refers to the place where research is conducted and by extension refers to an investigative approach and therefore to a research practice that is sometimes confused with the research object itself. It was not until the 1980s that the term fieldwork began to be deconstructed [9].

Given the polysemy of fieldwork, defining VFTs is not an easy task. Unlike fieldwork, VFTs constitute neither a research approach nor a place where one physically goes to collect data. A VFT is an investigative approach implemented in a space explored virtually using

digital tools. A VFT can lead to data collection. It is a learning approach that has the following characteristics [4]:

- It is a digital alternative to reality,
- It allows the exploration of space without being at the actual site,
- It is based on an interactive digital environment.

VFTs are based on active pedagogies [22], taking a constructivist [23] or social-constructivist perspective. This definition distinguishes between virtual visits, guided visits, and open VFTs.

### 2.3. Different Forms of Virtual Field Trips

Friess et al. [1] proposed a way to distinguish the different forms of field trips in an ordered plane based on two axes. The y-axis indicates the degree of student autonomy in the proposed tasks, while the x-axis indicates student involvement in the task. Figure 1 below allows us to visualize the differences between virtual visits, guided VFTs, and open VFTs.

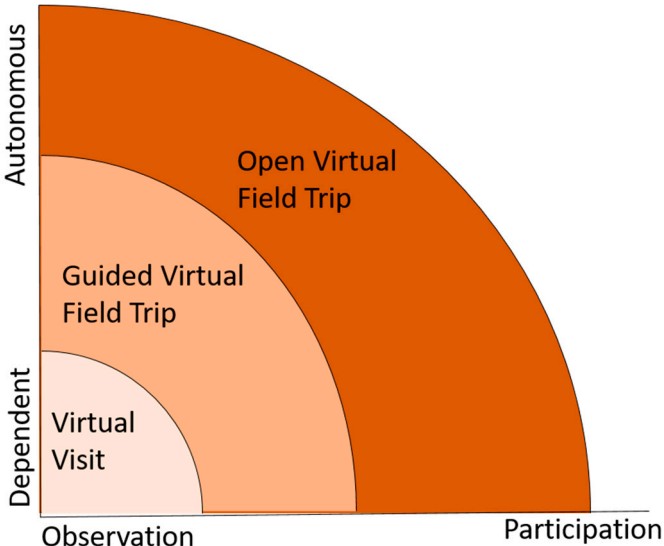

**Figure 1.** Typology of a virtual way of exploring a space (Own illustration adapted from [1]).

With a virtual visit, students observe and watch what others have created, such as a video or a presentation. They do not have the capability of moving around the space by themselves. Interactivity is limited, as is their autonomy, and students are dependent on a given tool. A guided VFT, however, is designed to be partly interactive. Students can freely manipulate specific tools to explore a portion of the space. For example, they can move around in a 360-degree photo or manipulate an interactive map. The tasks they are given are not limited to observation. A VFT is more participative than a virtual visit, and in this case, students are more autonomous. Geographical reasoning is guided by the tool and the task students must complete.

A VFT is not a tool but a learning process in which students freely explore space using various available tools. They use these tools in a complementary way to collect and produce data. With a VFT, geographical reasoning is opened. Students must choose the tools they will use and use them to collect data. They oversee all the geographical reasoning processes.

A VFT can be carried out before fieldwork to prepare for it or afterward to collect more data. The VFT can be produced after fieldwork [24]. It can also be an alternative to fieldwork [25]. VFTs can be a new way of being in the field using new technologies to collect data. France and Higgit [26] describe the use of these new technologies (e.g., GIS,

GPS, Google Earth) as "technologically enhanced" in their typologies of different forms of fielding.

VFTs can be mobilized asynchronously or synchronously. The tool does not constrain the use that can be made of it. For example, whatever the type of virtual field, it can be used in the classroom or as part of a blended learning or flipped classroom. As part of the Virt-Ex project, students made together virtual and real field work. Table 1 shows the design of this project. From this data, they created a VFT.

**Table 1.** The seminar design.

| Step | Description | Aims |
|---|---|---|
| Immersion | Some undertook sensitive fieldwork of a space identified as having an SD issue. Others explored these spaces virtually. The spatial practice at stake is that of the field. The diversity of practices results from the diversity of the places but also the diversity of the means of exploration (real visit/virtual field trip). | Have different experiences of places<br>Have spaces with different SD issues<br>Collect material, such as photos, videos, sensations, etc. |
| Interaction | Discussions are structured around several questions and instructions.<br>Conclude by comparing your various experiences.<br>What impressions did you have as you walked through this space? | Compare different experiences of the same space. |
| Institutionalization | Lectures of authors<br>Creation of mind map<br>Creation of Storymap | Conceptualize what a sustainable city is.<br>Theorize an approach to the use and production of VFTs. |
| Implementation | Using VFTs in the classroom | Develop a reflective practice on the use of VFTs in secondary schools. |

*2.4. Tools for Virtual Collaborative Fieldwork*

Many tools are available for conducting VFTs and creating guided VFTs. A teacher can choose specific tools depending on their intended goal [27], difficulty level, and digital abilities. In the context of VFTs, the following tools can be interesting.

2.4.1. First Virtual Exploration of Places

To explore any place on earth simply in two or three dimensions, easily accessible tools such as Google Earth [28] or Open Street Map are good choices. These tools can be applied in different geographical and educational contexts [29–31]. In the web version, no registration is necessary, so one can start right away. Using the search function, one can enter a location (e.g., HafenCity in Hamburg), which will then be searched and approached by the application. The area is marked on the map, and one can zoom in or out. In addition, one can walk along the streets of the location using Street View. This tool is ideal for getting first impressions about a place.

The National Aeronautics and Space Administration [32] site is also helpful for exploring places. However, it mainly contains images and animations. There are collaborative tools that have similar functionalities as Open Street Map and Geoportal. Geoportal does not allow one to walk around in 3D images but is a graphical interface with a GIS that allows one to explore the space through a wide range of layers.

2.4.2. Explore Already Existing Virtual Field Trips

In addition to the simple exploration of places, it is possible to make use of already existing field trips. There are already several available for different geographic topics.

Ready-made guided VFTs are available for physical geography topics (e.g., glaciers of the earth or cherry blossoms in Japan) or human geography topics (e.g., megacities or air pollution in London). They are available at all scales, from global to local (e.g., exploring Quebec or Singapore). In most cases, these are outdoor visits; indoor visits, such as tours inside factories, are rarer. These VFTs can take very different forms. Some are more or less interactive case studies, such as the ones in [33]. Some guided VFTs are more than just a documented analysis, and others are simply virtual visits, such as a virtual visit to the catacombs of Paris. Cultural institutions or cities often produce VFTs from a tourist perspective. Some VFTs are combinations of maps, accompanying text, images, and multimedia content. The open VFT entitled "Bogs: Witnesses of the Ice Age: A virtual field trip to the upper bog near Papenburg" is a mix of aerial videos, photos, exercises, and slide shows. This type of VFT is more recent. The differences between these different types of VFTs can be explained based on when they were produced and their purpose. These are available on the Google Earth website [28]. On that site, there are various geographical topics and quizzes. Another way to explore the world from home is offered by Esri Story Maps [34].

A wide variety of VFTs is available. Some universities offer platforms where one can find VFTs. In addition, there are other examples developed by individuals, often teachers, that are publicly available. Here is an example of a VFT to a bog in northern Germany: https://storymaps.arcgis.com/stories/d129b01ea71c4378b8dc25878465c441 (accessed on 4 March 2022). At the European level is the V-global project that develops VFTs in the context of global change (https://v-global.eu/) (accessed on 15 May 2022).

### 2.4.3. Digital Tools for Marking Places

Other exciting tools exist, such as map programs that allow one to mark places digitally with pins on a map. There are different programs available. For example, the map function of Padlet software is straightforward (https://de.padlet.com/) (accessed on 20 November 2021).

Padlet is not a geographic tool but software used to create a digital pinboard for collaborative work [35]. To date, it has mainly been used in disciplines other than geography [35,36]. An accurate map program is the software uMap (https://umap.openstreetmap.fr/de/) (version 1.2.3; France). uMap is open-source software that lets a user create a map relatively quickly.

### 2.4.4. Creating Virtual Field Trips

Another possible use of digital tools is to create VFTs. This is especially important for pre-service or in-service teachers, who can make these field trips either for students or with students. Various tools are available for this purpose. In addition to creating regular websites, one can use software that allows a relatively well-structured and straightforward presentation, such as ArcGIS Storymaps and Google Earth [28]. In addition to the above variants (see Section 2.3.), with ArcGIS Storymaps locations can be marked on a map and linked with different text, pictures, or videos to create a tour. Kerski [27] provides an excellent introduction to ArcGIS Storymaps and its possible applications.

## 3. Aims and Conceptualization of the Seminar

### 3.1. Aims of the Seminar

The objectives of the seminar were as follows.

*Research objectives*: The seminar aimed to explore virtual fieldwork in geography education. This exploration includes analyzing the advantages, disadvantages, opportunities, and limitations of real and virtual fieldwork. Of specific interest was which geographical working methods were suitable for virtual fieldwork. In addition to the options associated with the virtual format, the concerns and limitations were also of interest.

*Curriculum objectives:* Another goal was to develop a collaborative virtual teaching concept for virtual fieldwork in teacher education. This teaching concept should be flexible

enough to be integrated into different seminars and lectures on geography education at the two universities, Hamburg and Paris. This teaching concept would include the basics of virtual collaboration and digital map work and the development of VFTs.

*Skills objectives:* The seminar also aimed to develop the skills of students and teachers throughout the project. This was achieved through the reciprocal expertise available at the two universities. The focus was mainly on skills related to digital map work (see Section 2.4). Applications to facilitate collaborative working were also of interest.

*Content objectives:* In terms of content, the focus was on the topic of sustainable urban development. Specifically, this included various aspects of urban development, such as the city of the future, housing (neighborhoods as living space, linking the living and working, sustainable living), mobility (the city and the car, sustainable mobility, the smart city), and open spaces (urban gardens).

*ESD objectives:* Closely related to sustainable urban development is ESD, the guiding concept of the seminar. As the German and French educational systems differ, the guiding idea of ESD was compared and discussed against the background of the two systems. Afterward, the content (of sustainable urban development) associated with the ESD concept was discussed more concretely.

### 3.2. Organization of the Seminar

The seminar was built on different principles, as follows.

*A blended learning approach:* First, the seminar has a blended learning approach. Some activities were planned in real places. Real field trips concerning sustainability were designed in Hamburg and Paris—HafenCity as an example of urban development; Inselparkquartier as an example of sustainable housing; Science City Bahrenfeld as an example of a new science district; the Paris Rive Gauche district as a place of urban renewal to meet the challenges of densification in Paris; and Ecoquartier Bois Badeau (Brétigny-sur-Orge) as an example of urban development following the principles of sustainable development. Real field trips were used to collect data (images, videos) for the creation of Storymaps. Some activities occurred in a virtual learning environment (online conference, virtual workshop, online student presentation). Some sessions were also hybrid, with a face-to-face part and a remote part at the same time before the German and French lockdowns. The details are given in Table 1.

*Implementation in the curricula:* Second, the seminar was implemented in both the German and French curricula. The project involves seminars in geography education at both universities. The German students were in the Bachelor of Education program, while the French students were teachers involved in a master's program to improve their didactic skills. Third, the seminar is multicultural. It was designed so that both German and French students can compare and discuss their national educational systems (primarily how ESD is implemented in the curriculum), their living environment, their fieldwork practices, and how French and German geographers think about sustainable cities. Finally, the seminar was designed to be experiential.

*Experiential learning:* The experiential learning theory was formalized by Leon Kolb [37] in keeping with the ideas of Dewey [22], Lewin [38], and Piaget [23]. Dewey was the first to highlight the role of experience in learning. He identified two principles of an education for and through experience. He showed that all experience is part of the continuity of previous experiences. Not all experience is a source of learning but becomes so in exchanges with peers. Lewin [38] used this thesis in his work on changes in behavior within a social group. He developed the "unfreeze-change-refreeze" model, which consists of deconstructing habits (thawing) through discussion, which leads to changes in behavior (change). The new behaviors are then refrozen (regel) to be sustainable. Piaget has shown that experience does not have the same role in the construction of physical knowledge as in the construction of logical–mathematical knowledge. The former stems from empiricism, manipulation, and observation of objects. Logical–mathematical knowledge results from the abstraction of an action performed by the subject—classifying, sorting, ranking, etc. It is free of empiricism.

The process of knowledge starts with the confusion of the subject/object and moves towards increasing differentiation in a double movement of internalization (elaboration of the structure of knowledge) and externalization (objectivation of the object). It is based on this distinction that Piaget identified the different stages of a child's development. Kolb's theory of experiential learning starts with the observation that we live in an increasingly artificial world: "Our species long ago left the harmony of a non-reflective union with the 'natural' order to embark on an adaptive journey of its own choosing. With this choosing has come responsibility for a world that is increasingly of our own creation" [37]. In this artificialized world, we have lost the experiential dimension that is necessary for any learning process, even though knowledge has become an increasingly important issue in our societies. "We have lost touch with our own experience as the source of personal learning and development and, in the process, lost that experiential centered necessary to counterbalance the loss of scientific centered that has been progressively slipping away since Copernicus" [37].

Kolb developed a learning model to train young graduates in line with the socio-professional environment. Kolb [37] starts with the idea that a concrete experience confronts the subject with "divergent knowledge". The distancing of this knowledge through reflective observation then allows the subject to begin the process of assimilating this new knowledge, which leads him to conceptualize it. The conceptualization brings coherence to the knowledge that becomes "convergent".

Experiential learning is aligned with a holistic approach, which considers the person in every dimension—intellectual, ethical, psychological, and cultural. It promotes active learning, which places the learner and their experience at the heart of the learning process. This theory is founded on the premise that individuals can learn from their experiences by critically analyzing them, which leads them to conceptualize the experience. They can then assess the solidity and validity of their theoretical construction by testing it in experiments. Geographers, especially Anglo-Saxon ones, have adopted the experiential learning theory to develop experiential geography. This is geography education based on students' experiences, and it enables students to question their representations and spatial practices and to rethink them considering the knowledge and skills acquired in class [25,39–42]. However, the different steps of the approach designed by Kolb have not been adapted to the specificities. This is what the 4Is approach (Figure 2) designed by Leininger-Frézal and the Spatial Thinking research group (University of Paris) proposes [43]. The 4IS approach is an experiential geography model. Each step of the experiential approach is thought out in terms of the specificity of the knowledge, tools, and reasoning specific to geography.

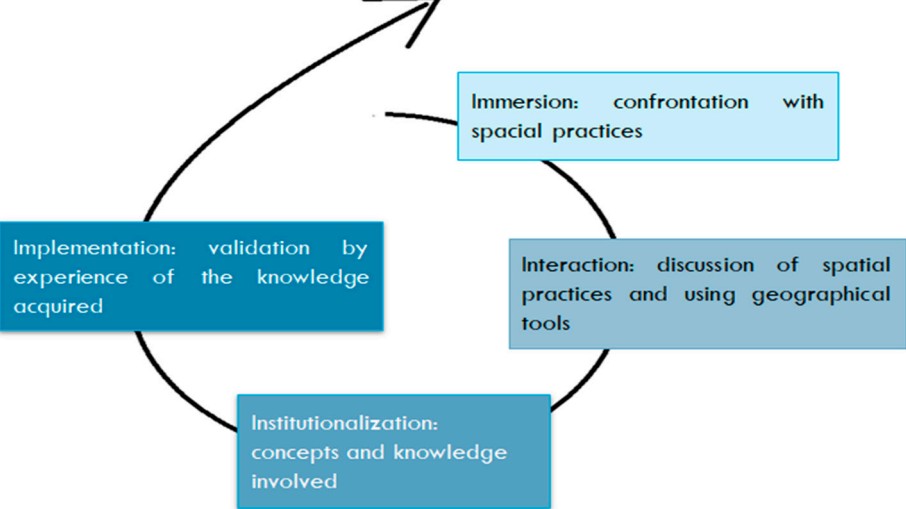

**Figure 2.** The 4Is approach (Source: Leininger-Frézal [43]).

Immersion is the phase in which the student is confronted with a spatial experience. To make an experience in space is not sufficient to speak about the spatial experience. Otherwise, any experience at all would be spatial. An experience is spatial if it is centered on spatial practice. Spatial practice is "the whole of the material and ideal relations of the individuals to the geographical space" [44]. Spatial practices are therefore real or ideal as representations. Moreover, the experience can be directly based on the students' own experience, or indirect when they report on the spatial practices of others.

Interaction requires exchanges between peers so that students can compare their experiences. They are thus led to compare, question, and analyze the spatial practices in play. As part of the interaction, students are encouraged to broaden and deepen their thinking by using geographic tools—mind maps (mainly thematic maps but not excluding topographic maps), plans, text extracts, GIS information, etc. Interaction is a resolutely socio-constructivist phase in which students are led to construct the concepts and notions at stake in the geographical situation studied. Interaction corresponds to the reflection phase in Kolb's [37] initial model.

Institutionalization refers to the formalization of the knowledge in play, namely the concepts and understanding of geography. It is a phase of conceptualization where students move from accumulating observations to arriving at an organized whole that makes sense. Institutionalization is essential for student learning. Unlike conceptualization, as defined by Kolb [37], institutionalization takes place within an academic discipline from which it cannot be freed, which influences its implementation.

The implementation is the moment of reinvestment of the learning achieved. This reinvestment can take place in class as part of an evaluation or another sequence. Implementation can also occur outside the classroom when the student realizes the integrity and operability of the knowledge learned in class in their everyday life. The following table shows how the 4Is approach is implemented in the seminar.

The immersion (first step of the 4Is approach) was asynchronous. Each student explored neighborhoods in their city according to the rules during the lockdown. Then, both interaction and institutionalization alternated between asynchronous collaborative group work and synchronous online or in-class exchanges. This teaching concept was initially developed for teacher training. However, they are to be very flexible so that they can also be used for students whose situation hinders physical mobility. Furthermore, it can be used in schools, for example, for the exchange of geography lessons with partner schools or homeschooling formats.

The organization of this seminar created several difficulties. Initially, the project had been designed with two field trips, one to Paris (France) and one to Hamburg (Germany). However, the maintenance of health measures until July 2021 made these trips impossible. It was decided to keep the project entirely online. The choice to work in the students' neighborhoods was also imposed by the traffic restrictions in the two countries (e.g., 3 km around the home in France during the different confinements). The exchanges being entirely online and in English was also a challenge for non-bilingual students. Their motivation and interest in the project were an asset to continuing the project.

An evaluation of the seminar was conducted in the empirical survey. As it is also of particular interest for geographical learning to relate to real and virtual formats, both variants were compared. Our research questions are as follows:

1. Is the 4Is model relevant for VFTs?
2. What geographical working methods are suitable for VFTs?
3. What opportunities and concerns relating to VFTs exist?

## 4. Methods

### 4.1. Sample

A total of 22 students took part in a questionnaire, with half of them from Germany and half from France. Regarding the results, the number of cases may be lower because the corresponding question was not answered by all participants. All students attended a master's seminar in geography education. Table 2 contains details about the sample.

**Table 2.** Participants.

| Student Characteristics | Germany | France |
|---|---|---|
| Number of students involved | 11 | 11 |
| Gender | 11 females | 3 males and 8 females |
| Type of teacher training | Pre-service teacher training | In-service teacher training |
| Number of students involved | Master of Geography Education | Master of didactics |

The methodological steps are shown in Figure 3. These consist of data collection and analysis. The results are then presented in the following chapter (see Section 5).

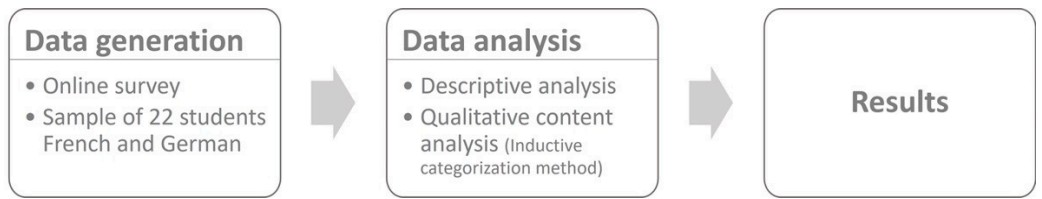

**Figure 3.** Methodological steps taken.

### 4.2. Data Generation (Questionaire)

An online questionnaire was conducted as part of this project. The questionnaire was developed based on existing literature on field trip education. The development of the closed-question items on geographical working methods was based on Lößner's items [45]. The questionnaire consisted of four closed questions (research questions 1 and 2) and three open questions (research questions 3 and 5). A five-point Likert scale was used for the closed-question items, ranging from "strongly disagree" to "strongly agree". The open-question items could be filled in freely by the participants. Quotation marks indicate the direct responses of the students presented in the results. The questionnaire was used after the seminar.

### 4.3. Data Analysis

To answer the research questions, we used descriptive analysis for the closed items and qualitative content analysis for the open items [46]. The evaluation follows the inductive categorization method, according to Mayring [46]. For this purpose, the content-related responses of the participants are divided into categories.

## 5. Results

This section presents the results of the evaluation of the seminar. Figure 4 shows learning opportunities in field trips. A distinction was made between real field trips and VFTs.

**Figure 4.** Learning opportunities of real and virtual field trips.

Figure 4 shows that real and VFTs are viewed differently by the respondents. The respondents think that real field trips have more advantages in various respects. Seven pre- and in-service students agreed that real field trips offer a more immediate confrontation with reality. Similarly, the majority felt that real field trips in particular can provide better knowledge of a place and practical exercises. Some students strongly disagreed that VFTs provide an "immediate confrontation with reality". Figure 5 shows how real and VFTs are seen by the participants regarding geographical working methods. Here, it becomes clear that participants' opinions of geographical working methods on real and VFTs differ. Two methods stand out where differences between real and VFTs can be observed. These methods are "Taking soil samples" and "Collecting items". Here, more than two-thirds of the participants disagreed or strongly disagreed. The four other methods (Figure 5), including "Drawing maps", "Drawing", and "Exploring in the field", can be better applied on real field trips. In further analyses, the open items were analyzed in terms of content. The students' quotations are placed in parentheses in each case.

**Figure 5.** Geographical working methods on real and virtual field trips.

### 5.1. Opportunities on Virtual Field Trips

Accessibility of field trip destinations: pre-and in-service students stated that VFTs allow for better accessibility of destinations "that you normally can't just visit". According to the respondents, VFTs allow students to reach destinations that are very far away ("Even more distant areas can be explored in a two-hour lesson"). Furthermore, VFTs may be conducted during school lockdowns (e.g., in the wake of the COVID-19 pandemic). In addition, VFTs can be helpful when there is no permission to conduct field trips. One respondent said, "I think that a virtual field trip in class can be practical because, in France, we don't always have permission to go outside. It is still not common to take your students outside for classes, at least in high school." Starting from the students' neighborhoods allowed the students to build geographical and professional knowledge on known spaces— geographical knowledge because it allowed them to question the stakes of sustainability in their environment and professional knowledge, as it allowed them to explore a learning approach linking geography and ICT. The 4Is approach allowed students to develop their skills based on their spatial experiences.

Inclusion: VFTs can help foster inclusion, as students can visit places they normally cannot for health reasons. This is where "read-aloud apps" should be used, for example.

Time factor: Unlike real field trips, VFTs can be better "integrated into lessons, especially when limited class time is available". Therefore, it is mentioned that it "can be used during a class session". In addition, if the field trip is already available, "less preparation time" is required and, unlike real field trips, no long-term planning is required; you can go "when you want".

Illustrativeness: A VFT is more illustrative than no field trip at all. If a real one cannot be conducted, a VFT "allows learners to get in touch with reality".

Digital skills: From the students' perspective, VFTs can "promote competencies in the use of digital media." This can also be associated with "high motivation" among students.

### 5.2. Concerns about Virtual Field Trips

No direct confrontation with reality: pre-and in-service students agree that only indirect confrontation with reality is possible in VFTs and that "the students cannot fully experience the space".

Lack of self-determination: pre-and in-service students also think that VFTs do not foster self-determination; there is "too less self-determination for students."

Inadequate technical equipment: pre-and in-service students raised concerns regarding technical equipment and devices. According to them, there are "Too few digital devices at school or in the home", and some students have an "insufficient internet connection".

Low willingness to engage: Some students raised the possibility that there might be poor engagement on the part of students with VFTs.

Information overload: Concerns were also expressed about possible information overload ("The students could be overwhelmed by the information overload").

### 5.3. Aspects of an Excellent Virtual Field Trip

This section presents the opinions of pre- and in-service students about what constitutes an excellent VFT. From their responses, it appears that several elements make a perfect VFT.

Interesting questions: Students frequently emphasized the necessity of an overarching "guiding question". An excellent VFT should therefore contain specific questions the students can use as a guide.

Interactivity: The pre- and in-service students repeatedly emphasized the aspect of interactivity. For them, an excellent VFT should employ interactive methods. A perfect VFT is "not just a digital guided tour but contains aspects for discovery". It must be "a program with movement and not a static map with photos".

Multiple methods: Closely related to interactivity is using various teaching techniques in a VFT. Students approve the "use of different media (maps, photos, videos, texts, worksheets)" in a VFT.

Structuring: The pre- and in-service students mentioned that good structuring of a field trip is appreciated, especially a "didactic pre-structuring."

Good preparation: From the pre- and in-service students' viewpoint, an excellent VFT should be well prepared. For them, a well-prepared VFT should include a final "presentation."

Accessibility: Relating to the idea of inclusion, the pre- and in-service students emphasized that VFTs should support "accessibility" by using, for example, apps where the content can be read aloud.

### 5.4. Substitutability of Real Field Trips with Virtual Ones

In a final question, pre- and in-service students were asked for their opinion on whether VFTs can replace real ones. Figure 6 shows the opinions of the respondents. An additional explanation is given in the text.

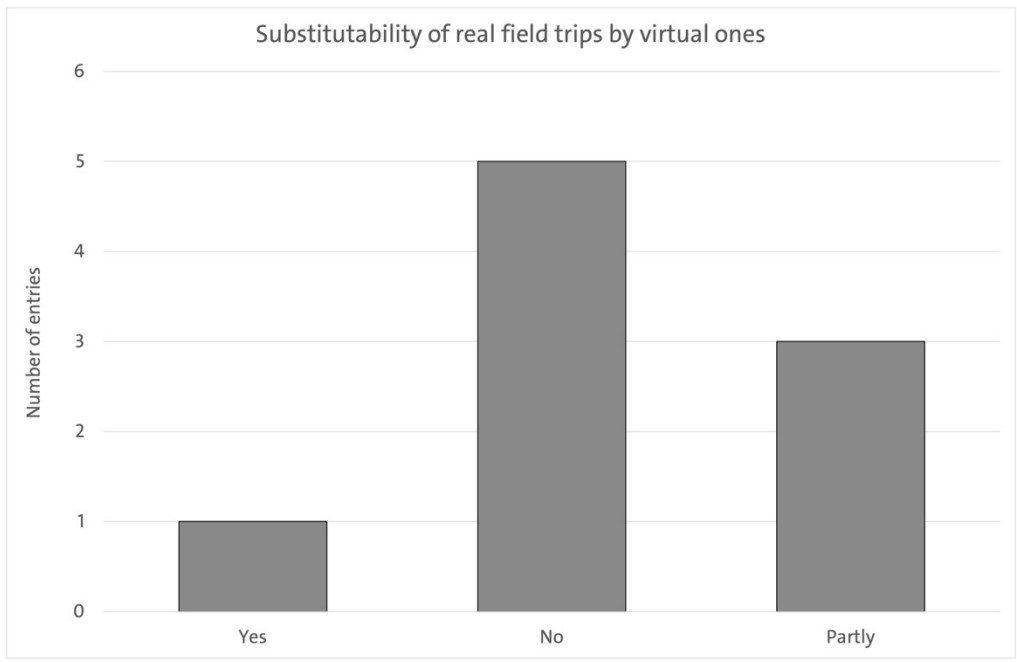

**Figure 6.** Substitutability of real field trips by virtual ones.

In one case, the question was answered in the affirmative, with the condition that students are allowed to use digital tools. In five cases, the question was answered in the negative. These students stated that this was for the following reasons:

Lack of interculturality: "All the cultural discovery is gone".

Limitations in geographic working methods: Limitations regarding geographic working methods were cited several times by pre- and in-service students. For example, one stated that no "experiments can be carried out." Another stated that "unfortunately, soil samples cannot be taken in the context of virtual field trips".

Inferior capacity to memorize the content of a VFT: Students mentioned that "real field trips are more likely to be remembered due to the experience with all/more senses on-site".

Limitations with the five senses: "You don't necessarily have the 3D dimension, nor do you have all the senses associated with visiting a space. For example, sound recordings don't reflect reality; you don't have the smell or the atmosphere of a place".

Three students think that real field trips can be partially substituted with virtual ones. They gave the following reasons for this:

Dependence on the question: Respondents believe that substitutability is possible, but substitutability depends "on the question of the field trip".

Time factor: VFTs are seen as good support for teaching when "real field trips cannot be carried out due to the time involved".

No real field trip possible: Several times, the students expressed that a VFT can be helpful if an actual field trip is not possible due to external conditions, for example, in the case of a "school lockdown due to the Corona pandemic".

Long distance from the field trip location: Students expressed that if a potential field trip location is very far away, a VFT can be a good alternative: "On the other hand, the use of VFTs can also make it possible to experience spaces that cannot otherwise be viewed more closely within the teaching setting. This is the case, for example, when a location is extremely far away. Therefore, in my eyes, both formats would ideally complement each other".

## 6. Discussion

This article has highlighted several aspects that extend the discussion on using VFTs in teaching geography in higher education. The results show that students see many learning opportunities during VFTs, as is also stated in the literature [5]. The first two research questions addressed comparisons between experience and geographical working methods on a real field trip versus a VFT. According to the students, real field trips have more significant advantages. They are more advantageous because they employ work methods that require direct contact with the physical world, such as taking soil samples or collecting plants. So far, there is no empirical evidence about this comparison, and the present study provides the first observations regarding this matter. Meanwhile, VFTs were examined in more detail. The third research question asked about the opportunities and challenges encountered during a VFT. The students expressed that VFTs make destinations more accessible. Better accessibility with VFTs is also described in the literature [1,4]. Through VFTs, more students can be brought into contact with geographical content in other places. Moreover, students also see the benefits of VFTs in terms of time. VFTs can be conducted when convenient [1]. Intricately linked to accessibility is the idea of inclusion. Inclusion is mentioned explicitly by the students and more implicitly in the literature [4]. These authors of this paper argue that VFTs provide access mainly when financial or physical reasons prevent a field trip in the real world. Furthermore, these authors emphasize the potential for international collaboration. The students expressed concerns regarding inadequate technical equipment. Even though there is an increasing trend toward using digital technologies in the classroom [1], technical hurdles can still hinder teaching and learning. In addition to the students' comments, the literature also mentions safety concerns, which is a significant issue on field trips [1,7]. In addition, the literature mentions financial aspects [1,2]. Friess et al. [1] show that the money available for field trips is decreasing. At this point, virtual formats can be a helpful alternative before no field trips can take place at all.

The fourth research question asked what makes an excellent VFT. Both the students interviewed and the literature [2,7] agree that good preparation is necessary. This includes a presentation at the end.

At the end of the presentation, the students and the teachers identified four criteria for an excellent virtual field.

Criteria 1: It is immersive. The tools and activities provided allow students to explore the space freely by moving virtually within that space.

Criteria 2: It is interactive. The VFT allows students to manipulate the tools at their disposal and to perform activities. It is not just about reading documents or watching videos.

Criteria 3: The VFT allows students to identify the stakeholders involved and the territorial process past and present.

Criteria 4: The VFT is an overview that allows students to explore the studied space at different levels of scale.

The fifth research question asked pre- and in-service students whether VFTs can replace real ones. In accordance with the literature [4], the students stated that topics cannot be

experienced with all five senses. Students tried to transcribe their experiences in their story map by adding sounds to the images. The rendering is necessarily limited, especially in terms of sounds and smells. The experience is inevitably different. For example, with a Google street map, it is possible to move only by making virtual jumps. It is not possible to move slowly. Some spaces, such as military buildings, are not visible, and some are entirely inaccessible. Virtual and real field trips are complimentary. A VFT is a technologically enhanced field trip.

The limitations of the study are presented below. The study contributes to the empirical discussion on VFTs. In the qualitative analysis, the spectrum of challenges and opportunities was highlighted. Limitations are seen in researching the effectiveness of VFTs. So far, there have been hardly any empirical findings regarding VFTs; therefore, a first step has been taken here. For future work, intervention studies in experimental or quasi-experimental studies that examine individual aspects of virtual excursions would be particularly necessary. Comparisons in different phases of education or of differences between real and virtual excursions are conceivable here. Effectiveness analyses should be carried out, ideally with group comparisons. A discussion about successful conditions for approaches is hugely relevant in an educational setting characterized by differences between digital and real formats.

Based on the results of this study, some implications for teacher education are as follows:

- With the present concept of VFTs, it was possible to show how collaborative international cooperation can take place.
- Both variants, real field trips, and VFTs, have relevance. Their use depends on the individual setting (e.g., location of the educational institution, site of the field trip, group of participants and their needs, school situation, etc.).
- A VFT is a way to develop students' digital skills to help them learn using digital tools.

**Author Contributions:** Conceptualization, C.L.-F. and S.S.; methodology, C.L.-F. and S.S.; software, C.L.-F. and S.S.; validation, C.L.-F. and S.S.; formal analysis, C.L.-F. and S.S.; investigation, C.L.-F. and S.S.; resources, L.-F.C.; data curation, C.L.-F. and S.S.; writing—original draft preparation, C.L.-F. and S.S.; writing—review and editing, C.L.-F. and S.S.; visualization, C.L.-F. and S.S.; project administration, S.S.; funding acquisition, S.S. All authors have read and agreed to the published version of the manuscript.

**Funding:** The project on which this publication is based was funded by the German Federal Ministry of Education and Research (BMBF) and the German Academic Exchange Service (DAAD) within the funding program International Virtual Academic Collaboration (IVAC). Responsibility for the content of this publication lies with the author.

**Institutional Review Board Statement:** Not applicable.

**Informed Consent Statement:** Informed consent was obtained from all subjects involved in the study.

**Data Availability Statement:** Data available on request due to restrictions eg privacy or ethical.

**Conflicts of Interest:** The authors declare no conflict of interest.

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
