# Peer review of "Virtual Field Trips in Binational Collaborative Teacher Training: Opportunities and Challenges in the Context of Education for Sustainable Development"

_sustainability, doi:10.3390/su141912933_

Round 1
Reviewer 1 Report
This manuscript is about the evaluation of a virtual field trip. I am afraid that more work needs to be done in the introduction and methods section before the manuscript can be re-considered via another round of peer-review.
The authors mentioned in the introduction that their research is based on the 4I model, however no description of the model was given. In addition, the hypothesis of the research stated in the introduction does not seem to be related to the research questions listed at the end of section 3.2. Also, I find some of the research questions vague. For example, the “aspects” of interest in Research Question 1 were not introduced in the section preceding the presentation of the research question. Thus, it is unclear what the “aspects” are referring to.
In the method section, a sub-section on the instrument (i.e. the questionnaire) need to be added to provide readers with a description of the questionnaire, in terms of how the survey was developed, the number of items, the rating scale used etc.
Reviewer 2 Report
This article focuses on the possibilities and limitations of virtual tours in the framework of two-way collaborative teacher training. The article is interesting and informative. The present study represents a contribution to the development of sustainability issues.
The methodology is consistent with the goal. The design of the article and the conclusions clarify the task.
The authors have done a great job, but the article needs a little improvement.
In line 55. It is necessary to clarify what is meant by the word "It" "It is aimed at ensuring ...". From the context, of course, this meaning is clear, but the presence of clear sentences is characteristic of the scientific language.
Line 132. This is what 26 refers to (appends s) as,.." You also need to qualify "This".
It is necessary to indicate which tools are meant: Line 143 "tools such as [28] are a good choice." Moreover, the authors evaluate them positively.
Also, the authors need to highlight the issue of the existence of special projects that allow you to make excursions online.
The article is devoted to education for sustainable development, so the authors can touch on the topic of environmental projects. This aspect will strengthen the work on sustainability. The authors touched upon the topic of excursions in general, but it is necessary to show that there are also special projects. Moreover, this aspect of the work will expand the reference section. The authors need to expand it. To reveal the aspect of existing projects, the authors can familiarize themselves with the following works:
Brown, E.D., & Williams, B.K. (2019). The potential of citizen science to produce reliable and useful environmental information. Conservation Biology, 33, 561-569.
Shutaleva A., Nikonova Z., Savchenko I. and Martyushev N. (2020). Environmental education for sustainable development in Russia. Sustainability (Switzerland), 12(18), [7742]. https://doi.org/10.3390/su12187742
Kobori H., Dickinson J., Washitani I., Sakurai R., Amano T., Komatsu N., Kitamura V., Takagawa S., Koyama K., Ogawara T., and Miller-Rushing, A.J. (2015). Citizen science: a new approach to the development of ecology, education and conservation. Environmental Research, 31, 1-19. https://doi.org/10.1007/s11284-015-1314-y
Reviewer 3 Report
1. In the Introduction part, the organization of the paper must be reflected.
2. In 2.4, more types of virtual collaborative fieldwork may be discussed like Synchronous and Asynchronous concept may be induced.
3. In section 4, the method may be represented in a diagram form or flow chart form to make more transparent.
4. In results section, Figure[4-6] may described in theoretical aspect i.e. why this kind of results are generated.
5. The future works have not been discussed properly. it should be like more authors can work based on this concept paper.
6. References are not in proper format.
Reviewer 4 Report
I liked the paper and the topic. Thanks.
I feel the theoretical modelization proposed is not used and further discussed after or about the case study proposed. How did you develop the model proposed? How the case study help to develop it, and how the model was applied to the case study? How Kolb's model is used in the case study proposed?
Stress the difficulties and challenges of the international application of virtual fieldwork. Stress a little more about the collecting data, considering we have a sample of only 22 students. Why did you not proceed with some interviews?
In general, a more general coherence internal to the paper will be remarkable to make the paper clearer.
